# Molecular architecture and conservation of an immature human endogenous retrovirus

Anna-Sophia Krebs [1,8], Hsuan-Fu Liu [2,8], Ye Zhou [3,8], Juan S. Rey [4], Lev Levintov[4], Juan Shen[1], Andrew Howe [5], Juan R. Perilla [4] ✉, Alberto Bartesaghi[2,3,6] ✉ & Peijun Zhang [1,5,7] ✉

The human endogenous retrovirus K (HERV-K) is the most recently acquired endogenous retrovirus in the human genome and is activated and expressed in many cancers and amyotrophic lateral sclerosis. We present the immature HERV-K capsid structure at 3.2 Å resolution determined from native virus-like particles using cryo-electron tomography and subtomogram averaging. The structure shows a hexamer unit oligomerized through a 6-helix bundle, which is stabilized by a small molecule analogous to IP6 in immature HIV-1 capsid. The HERV-K immature lattice is assembled via highly conserved dimer and trimer interfaces, as detailed through all-atom molecular dynamics simulations and supported by mutational studies. A large conformational change mediated by the linker between the N-terminal and the C-terminal domains of CA occurs during HERV-K maturation. Comparison between HERV-K and other retroviral immature capsid structures reveals a highly conserved mechanism for the assembly and maturation of retroviruses across genera and evolutionary time.

When the human genome was first sequenced, it was found that ~8% can be classified into the broad class of human endogenous retroviruses (HERVs)[1]. These viruses integrated into the genome over millions of years and most have accumulated mutations rendering them non-infectious. HERV-K (HML-1 to HML-10) is the most recently incorporated and most active virus group, especially its subgroup HML-2. HERV-K is present with hundreds of copies in the genome and some HERV-K copies consist of a complete open reading frame with all structural and regulatory proteins present but none can complete a viral life cycle due to mutations in some proteins. While HERV-K is not expressed in healthy adult cells, some loci are upregulated in various cancers[2–6], in amyotrophic lateral sclerosis (ALS)[7], during HIV-1 infection[8], and most recently in organs and tissues of aged primates and mice[9,10]. When HERV-K copies are being expressed in these cells,

they can produce virus-like particles (VLPs) that are non-infectious. However, a consensus sequence based on ten full-length proviruses of HERV-K, termed KERV-K_con, was shown to produce infectious virions[11,12], whose life cycle is inhibited by a variety of restriction factors such as tetherin and APOBEC family members that restrict exogenous retroviruses. Therefore, HERV-K serves as a paradigm for the understanding of the architecture and property of endogenous retroviruses when integrated into the human genome thousands of years ago. They constitute a fossil record saved in our genomes and provide a conservation pattern that have evolved over millions of years of interplay between retroviruses and their hosts. Moreover, HERV-K is also of medical importance as HERV-K VLPs induce senescence in young cells[10] and a number of genes are under definite transcriptional control by HERV-K loci during tumorigenesis[13].

[1]Division of Structural Biology, Wellcome Trust Centre for Human Genetics, University of Oxford, Oxford OX3 7BN, UK. [2]Department of Biochemistry, Duke University School of Medicine, Durham, NC 27710, USA. [3]Department of Computer Science, Duke University, Durham, NC 27708, USA. [4]Department of Chemistry and Biochemistry, University of Delaware, Newark, DE 19716, USA. [5]Diamond Light Source, Harwell Science and Innovation Campus, Didcot OX11 0DE, UK. [6]Department of Electrical and Computer Engineering, Duke University, Durham, NC 27708, USA. [7]Chinese Academy of Medical Sciences Oxford Institute, University of Oxford, Oxford OX3 7BN, UK. [8]These authors contributed equally: Anna-Sophia Krebs, Hsuan-Fu Liu, Ye Zhou. ✉e-mail: jperilla@udel.edu; alberto.bartesaghi@duke.edu; peijun.zhang@strubi.ox.ac.uk

Phylogenetically HERV-K belongs into the betaretrovirus-like supergroup as its sequence is most similar to mouse mammary tumor virus (MMTV)[14,15]. The main structural protein Gag is sufficient for HERV-K to produce immature VLPs. Upon maturation Gag gets cleaved by its protease into matrix (MA, binds to the membrane), p15, capsid (CA, forms the inner core), nucleocapsid (NC, binds the RNA) and three spacer peptides (SP1, QP1 and QP2)[16,17]. The mature HERV-K capsid structure was recently elucidated by single particle cryoEM, using in vitro assemblies of recombinant HERV-K$_{con}$ capsid protein (CA)[18]. However, the immature HERV-K Gag structure and its assembly as an immature lattice remain elusive.

Here we present a cryoET study of the native immature HERV-K$_{con}$ Gag VLPs released from human cells. Interestingly, HERV-K$_{con}$ Gag VLPs are much larger than other known retroviruses, with a substantially larger gap between the membrane and the capsid layer. We used cryoET and subtomogram averaging (STA) to determine the structure of HERV-K CA domain of immature Gag at 3.2 Å resolution. Its overall structure is analogous to other retroviruses, demonstrating strong conservation between human endogenous and exogenous retroviruses. While the individual N-terminal (NTD) and C-terminal (CTD) domains of the capsid are structurally conserved between the mature and immature forms, the hinge between these two domains, however, is different, leading to vastly different NTD and CTD orientations and consequently distinct lattices for the mature and immature HERV-K. The presence of a conserved density at the top of the 6-helix bundle (6HB) of the immature CA hexamer, coordinated by two rings of conserved lysine residues, suggests a remarkably similar mechanism of inositol hexaphosphates (IP6) in HERV-K assembly and infection compared to IP6 in HIV-1 and other retroviruses such as EIAV and RSV[19–21].

## Results and discussion

### Purification and characterization of HERV-K Gag VLPs

To compare the morphology of HERV-K with exogenous viruses, we expressed HERV-K$_{con}$ (HERV-K thereafter for simplicity) VLPs in HEK293T cells and purified them with a final density gradient ultracentrifugation (Supplementary Fig. 1a). There were two VLP-containing bands in the gradient, which were then plunge frozen separately in liquid ethane for cryoET analysis. The upper band had many fused VLPs (Supplementary Fig. 1b, panel III), whereas the lower band consisted mostly of nicely formed individual VLPs suitable for structural analysis. These VLPs are spherical in shape, contain an outer lipid bilayer and an inner ring for the CA lattice (Fig. 1a–c). The diameter of the HERV-K VLPs ranges from 120 nm to 210 nm with a mean diameter of ~173 nm (Fig. 1e), significantly larger than other retroviruses, including HIV-1 and HIV-2, RSV, MLV, HTLV-1 and HFV[22]. Compared to the HIV-1 immature VLPs, there appears a wider gap between the lipid bilayer and the capsid lattice (Fig. 1c, d). We further measured the distance between the membrane and CA, resulting in a gap distance ranging from 16 nm to 24 nm with a mean value of 21 nm, much larger than the gap in HIV-1 (~10 nm)[22], but close to the gap in RSV (~18 nm)[22] (Fig. 1f).

The greater distance between the CA lattice and the membrane is most likely due to the presence of the segments SP1 and p15 between MA and CA. P15 encodes a late (L) domain motif which is needed for the recruitment of host proteins responsible for the release of virions from the plasma membrane[16,17]. HIV-1, HIV-2 and HTLV-1 have no extra domains between MA and CA. MMTV has 4 proteins (pp21, p3, p8, n) without a late domain but p8 and n are required for assembly[23]. M-PMV, RSV and MLV each encode additional segments between MA and CA (pp24(L)/p12; p2a/p2b(L)/p10(L); p12(L) respectively)[24–28], and

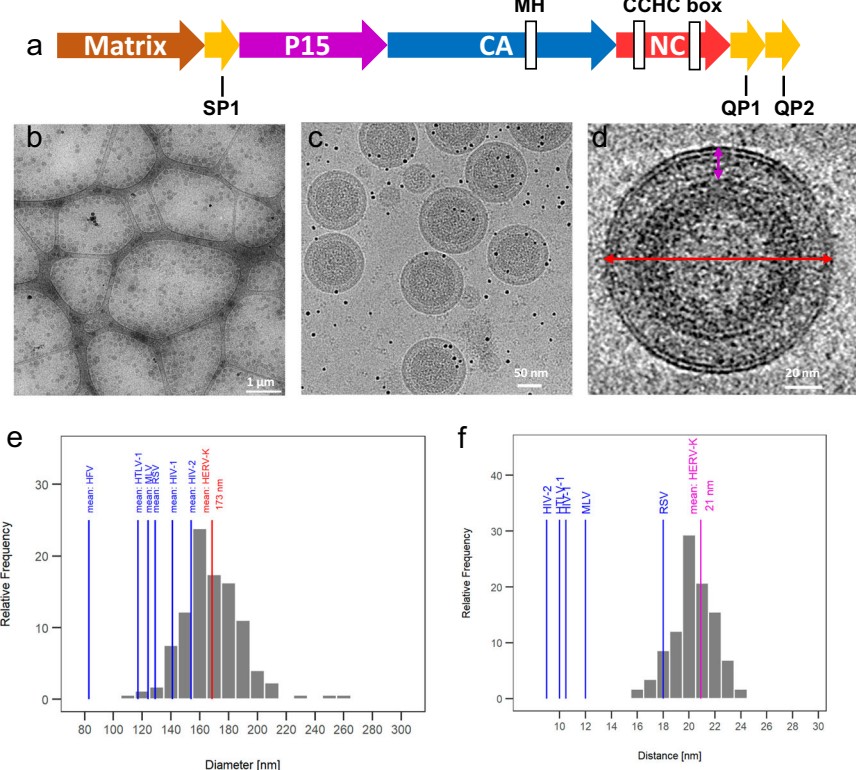

**Fig. 1 | CryoEM analysis of immature HERV-K virus-like particles (VLPs).**
**a** Schematic representation of Gag domains. **b**–**d** Representative cryoEM images of purified immature HERV-K VLPs at low (**b**), medium (**c**) and high (**d**) magnifications. **e** The mean (red line) and distribution (gray bars) of the diameter (red arrow in **d**) of HERV-K VLPs, compared with the mean diameters of other retroviruses (blue lines).

**f** The mean (purple line) and distribution (gray bars) of the distance between lipid bilayer and capsid lattice (purple in **d**) of HERV-K VLPs, compared with the mean distance of other retroviruses (blue lines). The micrographs in (**b**) and (**c**, **d**) are representative of 20 and 124 measurements, respectively. Source data are provided as a Source Data file.

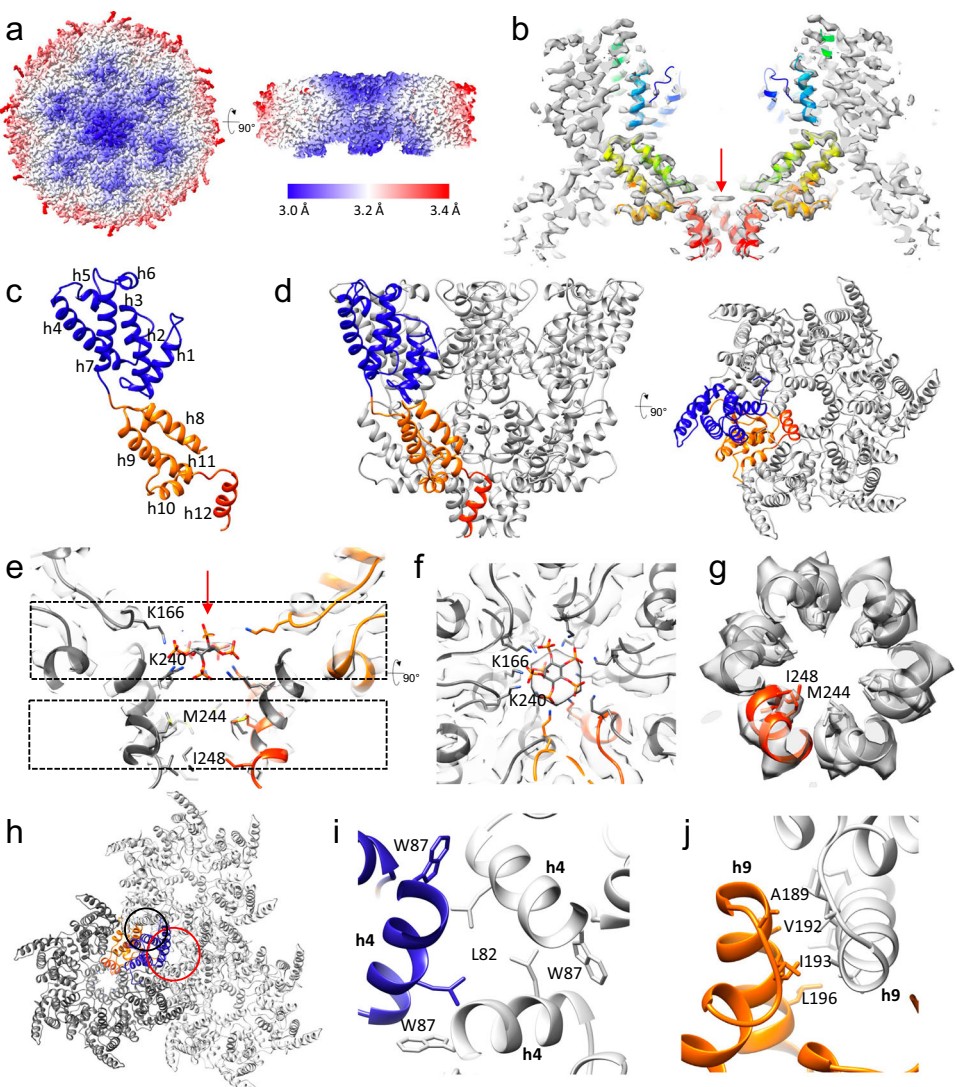

**Fig. 2 | CryoET STA of immature HERV-K VLPs. a** CryoET STA map of immature HERV-K CA hexamer colored by local resolution (red to blue). **b** A slab of HERV-K CA hexamer map overlapped with the atomic model (rainbow colors blue to red, N-terminus to C-terminus). **c** The atomic model of the immature HERV-K CA monomer (NTD in blue, CTD in orange, h12 in red) with α-helices labeled. **d** The atomic model of the immature HERV-K CA hexamer viewed from the side and top with one monomer colored in blue and orange. **e** A central slice of the HERV-K hexamer density map, showing IP6 density with an IP6 fitted (red arrow) and two Lys rings (K166 and K240), in addition to the hydrophobic core of the 6HB (M244 and I248). **f, g** Density slabs at the top (**f**, top dashed box in **e**) and bottom (**g**, bottom dashed box in **e**) of the 6HB, with the protruding side chains shown as sticks. **h** The dimer (black circle) and trimer (red circle) interfaces between adjacent immature HERV-K hexamers. **i** A detailed view of the trimer interface involving helix 4 with the contributing side chains shown as sticks. **j** A detailed view of the dimer interface involving helix 9 with the involved side chains shown as sticks.

consequently, display a larger gap compared to HIV-1[22,29]. The observed large distance between the membrane and the CA lattice in HERV-K correlates well with the presence of these additional segments between MA and CA.

## Structure of HERV-K immature Gag and lattice

The immature HERV-K VLPs are heterogeneous with variable sizes and morphologies (Fig. 1), thus we employed cryoET combined with STA to determine the structure of the immature CA lattice. The centroids of ~800 VLPs were computationally detected from the tomograms in an automated manner and used as seeds to find the location of the external surface of the CA lattice using unsupervised segmentation routines. 270 K uniformly distributed positions were identified along the CA lattice of all VLPs and sub-volumes with a side length of 40 nm were extracted for further processing (Supplementary Fig. 2a). Sub-tomogram averaging followed by analysis of the raw tilt-series projections using STA refinement procedures that impose the constraints

of the tilt-geometry, resulted in a 3.2 Å resolution map (Fig. 2a, Supplementary Fig. 2b). Mapping of individual subtomograms back to the original tomograms reveals that the refined subtomogram positions are arranged in a hexagonal lattice with incorporated gaps to accommodate the lattice curvature (Supplementary Fig. 3). This type of hexagonal immature CA lattice is seen in many other retroviruses and seems to be a conserved feature across retroviral genera. It thus likely has evolved earlier than the integration of HERV-K into the human genome. Subtomogram averaging could resolve the CA domain of Gag (Gag amino acid sequence 283–532) and no well-resolved density was observed for MA or NC. Sidechains in the STA map of HERV-K immature CA were clearly resolved (Fig. 2a, Supplementary Fig. 2b, d), thus allowing atomic model building (Fig. 2b). The CA monomer consists of an N-terminal domain with 7 helices and a C-terminal domain made up of helices 8-11 and a terminal helix 12 (Fig. 2c, d). The linker between the NTD and CTD is also well resolved. The first 21 amino acids at the N-terminus are flexible and were not included in the atomic model.

Like other retroviral immature CA structures, there are no NTD-CTD interaction interfaces present within the hexamer[29]. The CA monomers assemble into a hexamer through the terminal helices h12, forming the 6HB (Fig. 2d). At the top of the 6HB, there is a strong density, most likely corresponding to the 647.94 Da myo-inositol hexakisphosphate (IP6) molecule, coordinated by two rings of lysine residues, K166 and K240 (Fig. 2e, f), remarkably similar to the immature CA hexamer in HIV-1, where the IP6 molecule neutralizes the charges of two lysine rings, K158 and K227, thus stabilizing the 6HB[30]. Further downstream in the 6HB, hydrophobic residues at the inner face of the 6HB, facilitate the bundle formation (Fig. 2e, g). The assembly of the immature CA lattice is mainly mediated by the hydrophobic interactions at the trimer (N-terminal h4 helix) and the dimer (C-terminal h9 helix) interfaces between adjacent hexamers (Fig. 2h–j). In contrast the Mason–Pfizer monkey virus (M-PMV), the only immature structure solved in the betaretrovirus family, does not display the 6-helix bundle at the C-terminus or the N-terminal trimer interface[31]. However, the presence of M-PMV's CA-NC junction is still essential for assembly, maturation and infectivity[31]. As HERV-K assembles at the plasma membrane[2], unlike other betaretroviruses that assemble in the cytoplasm, the 6HB might be a feature that was lost in some betaretroviruses at some point during viral evolution.

## HERV-K immature Gag interface analysis using molecular dynamics simulations

Prior to conducting molecular dynamics simulations, we built and refined a structure model of the HERV-K CA hexamer via comparative modeling using RosettaCM[32] (Supplementary Fig. 4). The refined model was used to assemble and perform molecular dynamics simulations to probe the structural integrity and overall stability of interactions formed at the trimer and dimer interfaces of the HERV-K CA hexameric lattice (Supplementary Fig. 5) as well as to probe the intradomain interactions between the individual monomers. Therefore, an initial model amenable to atomistic simulations was refined from the density data as described in the methods section below and following a procedure previously developed by Dick et al.[33].

Heavy atom root mean square fluctuations (RMSF) were derived from the simulations to determine the flexibility of amino acids located at multiple CA interfaces. Interfacial residue interactions were also identified by measuring residue to residue distances from the simulation trajectory. Specifically, salt bridging interactions were characterized by the distance between interacting atoms while for hydrophobic interactions the center-of-mass to center-of-mass distance of the sidechain heavy atoms was measured. Flexibility is a good indicator of the importance and specificity of contacts as more rigid contacts tend to be more essential to stability of an interface, at the same time, more flexible residues tend to have more spread interaction distance distributions indicating fluctuating interactions and lower occupancies. For instance, we initially identified the R100 and Q118 (located in h4) to be the most flexible residues at the inter-hexamer trimer interface as characterized by the highest RMSF values (Fig. 3a–c). Due to this flexibility, Q118 periodically formed hydrogen-bonding interactions with R100 while also interacting with the solvent (Fig. 3a, Supplementary Fig. 6a). Furthermore, R100 also formed hydrophobic interactions with S121 and T122 (Fig. 3a, Supplementary Fig. 6a). We tested the effect of charge suppression by alanine mutation to these residues by measuring the efficiency of VLP production using western blots. These results were consistent with their ability to retain assembly when mutated and proving that the chemical nature of these residues is not essential for the formation of the trimeric interface (Fig. 3g).

Stable interactions were found at the inter-hexamer trimeric interface dominated by charged residues and complemented by hydrophobic residues located in h4. Hydrophobic interactions at the inter-hexamer trimer interface between L82 and W87 were shown to provide stabilization to the trimer interface (Fig. 3c, Supplementary Fig. 6c). Closer inspection revealed that L82 formed unspecific hydrophobic clusters with each other and W87 (Supplementary Fig. 6c). Charged residues R97, E69, K85 and D90 as well as hydrophobic residue Y66 were identified as not flexible according to their low RMSF values (Fig. 3b). Correspondingly, their density is well resolved in the cryo-ET map compared to flexible residues in the inter-hexamer trimer interface (Supplementary Fig. 7). We encountered that R97 formed salt bridging interactions with E69 and amino-aromatic interactions with Y66 (Fig. 3b, Supplementary Fig. 6b); and K85 formed salt bridging interaction with D90 (Fig. 3b, Supplementary Fig. 6b). Charge removal from each one of these residues separately, as well as changing aromatic residue, completely abolish VLP production (e.g., mutations to alanine) (Fig. 3g). The results confirm the stabilizing role of these amino acids at the inter-hexamer trimer interface in immature HERV-K Gag assembly.

At the inter-hexamer dimer interface we could confirm hydrophobic interactions between h9, similar to other retroviruses. On average, residues at the inter-hexamer dimer interface were less flexible in comparison to residues at the inter-hexamer trimer interface with the exception of L196 (Fig. 3d, Supplementary Fig. 6d). In addition, V192 interacted with L196 and periodically with I193 or M197 (Supplementary Fig. 6e), showing possible partial occupancies of V192. We also observed that E135 formed salt bridging interactions with R98 (Supplementary Fig. 6f). Furthermore, we observed the formation of salt bridges between the E132 of neighboring hexamers mediated by sodium ions (Supplementary Fig. 6g), in agreement with non-protein density regions observed in the cryo-ET map in proximity to the E132 residues (Supplementary Fig. 8).

When monitoring the side-chain flexibility at the intrahexameric interactions, we confirmed the poor stability of hydrophobic interactions between rings of M244 and I248 at the 6HB (Supplementary Fig. 6h). This lack of stability was captured by their increased flexibility in comparison to K166 and K240 (Fig. 3e, f). Addition of IP6 molecules to the cavity of the hexameric rings, stabilized both K166 and K240 rings through the formation of salt bridging interactions (Supplementary Fig. 6i). The K166 and K240 rings were observed to be more flexible in the absence of the IP6 molecules as reflected by increased root mean square deviations (RMSD) in comparison to the hexamers with bound IP6 molecules (Fig. 3e, f, Supplementary Fig. 9). In addition, the IP6 molecule is found to be stable in the 6HB, as measured by a low RMSF in the carbon ring atoms while the oxygens in the phosphoryl groups are the most flexible (Supplementary Fig. 10), in accordance with the observed density.

## Comparison between immature and mature HERV-K CA

When comparing the immature form of HERV-K CA with its mature counterpart[18], it shows clearly that the NTD and CTD separately overlap almost perfectly and the number of helices and their orientations in the separate domains does not change, except for the addition of h12 in the immature structure (Fig. 4a) (note that the numbering of helices differs from Acton et al.[18] to comply with other retroviruses where helix 9 forms the CTD-CTD interface). However, the hinge between the domains in the immature CA adopts a completely different conformation from the mature form, resulting in a different orientation between the NTD and the CTD, and consequently neither the dimeric nor trimeric interfaces are alike. This is analogous to the conformational difference observed between the mature and the immature CA in HIV-1[19].

In the immature lattice, the NTD and CTD are located on top of each other which makes it impossible to form interactions between the two domains. This gives it a tighter, more vertically elongated packing scheme than the mature lattice (Fig. 4d, e). The tight packing allows the virus to form a completely new set of interactions in its lattice, including the 6HB (Fig. 4b), the dimer interface and the trimer

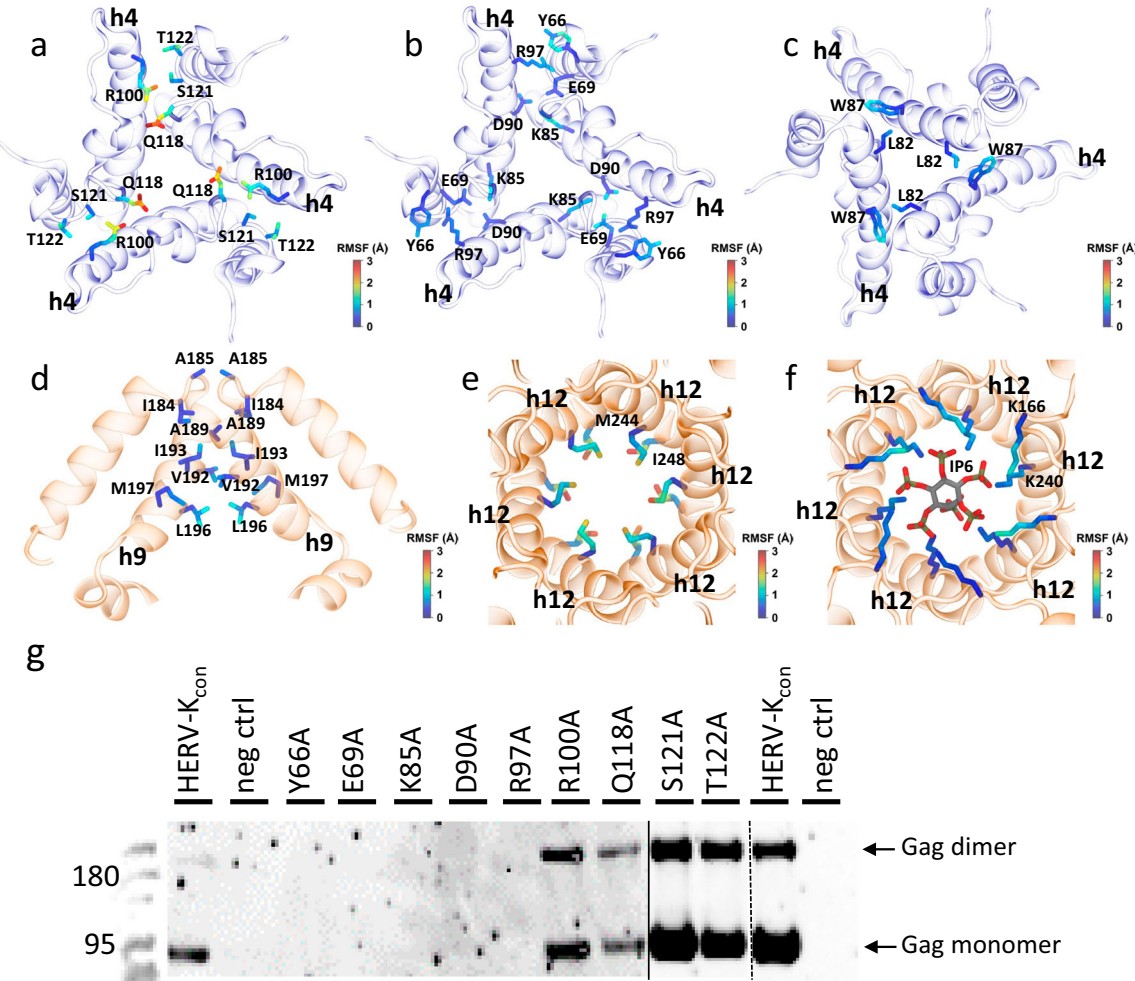

**Fig. 3 | MD simulations and mutational validation of immature HERV-K CA intermolecular interactions. a–c** Snapshots of the inter-hexamer trimer interface involving helix 4 with the contributing side chains shown as sticks. Residues in (**b**) and (**c**) show stable interactions. **d** A snapshot of the inter-hexamer dimer interface involving helix 9 with the contributing side chains. **e, f** Views of the 6HB with the contributing side chains of the hydrophobic core (M244 and I248) (**e**) and the K166 and K240 rings with the IP6 molecule (**f**). In each snapshot, protein is represented as cartoon (NTD in blue, CTD in orange). The amino acid side chains are colored based on the heavy-atom RMSF computed throughout the MD simulation. **g** Western-blot of VLP assembly carrying trimer interface mutants of HERV-K Gag. The Western-blot experiments, including independent infections, were repeated three times. Source data for (**g**) are provided as a Source Data file.

interface (Fig. 4b–e). The dimer interface in immature HERV-K is very similar to M-PMV, yet more extensive compared to HIV-1, especially involving the NTD[19]. Like HIV-1, the HERV-K dimeric CTD-CTD interactions are formed by helix 9 in both the immature and mature structure. While some of the interacting side chains, A189, V192 and I193, are preserved[18], the crossing angles of the h9, however, are very different (−50° immature vs. 95° mature), resulting in a different interface overall. The similarity between HERV-K and HIV-1 in the immature interfaces and in the conformational difference between the mature and the immature CA suggests a likely conserved structural maturation process[34,35].

**Structure conservation in retroviruses**

Structurally, the arrangement of CA domains in intact immature HERV-K is most similar to the immature M-PMV and RSV but deviates from immature HIV-1 and MLV (Fig. 5a). Such difference in CA domain orientation has been previous noticed between M-PMV and HIV-1[19]. When comparing HERV-K to HIV-1, the sequences of the terminal helices in HIV-1 and HERV-K are strikingly similar (Supplementary Fig. 11). HERV-K and HIV-1 both bind IP6 by two lysine rings[19] (Fig. 5d). RSV, an alpharetrovirus, binds IP6 in a similar pocket, but via one ring of lysines and one ring of arginines[20]. While

HERV-K, HIV-1, MLV and M-PMV all use their helix 9 to form a dimeric interface in the CTD, MLV forms additional interactions between its 3/10 helix and the base of h9[36]. Other retroviruses use additional dimeric interfaces in the NTD, such that MLV involves helix 7 and the loop between helix 4 and 5[36], RSV forms its dimeric interface using helices 2 and 7 and needs its upstream region p10 to form the CA lattice[37], whereas HIV-1 requires H1/H2 to form the dimeric/trimeric interfaces in the NTD[29].

A structure-based phylogenetic tree can be used to sort protein models by their similarity without taking into account their amino acid sequence. This allows for comparisons which are not biased on how closely related two viruses are. A tree based on published immature structures allows us to see similarities not otherwise obvious (Fig. 5c). A comparison with the mature HERV-K CA can explain why it is located furthest away from its immature form, even though it obviously has the same amino acid sequence as its immature form.

Surprisingly though, the immature betaretrovirus, M-PMV, is not situated next to the immature HERV-K model. It has a similar NTD as HERV-K, which can be seen in their similarity of the trimeric interface. However, its helices in the CTD differ significantly; and, as already discussed, it lacks the h12 helix at the C-terminus which forms the 6HB in the other retroviruses (Fig. 5b). It is worth mentioning that MPMV is

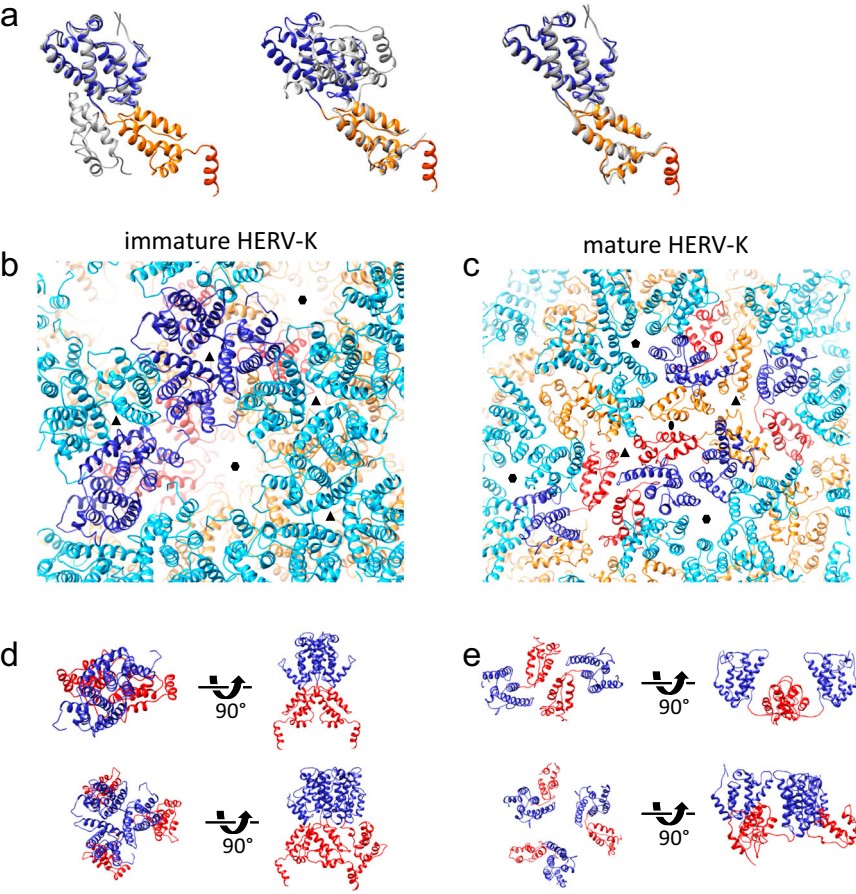

**Fig. 4 | Comparison between immature and mature HERV-K CA. a** Overlay of the immature HERV-K structure (NTD blue, CTD orange, h12 red) with mature HERV-K (gray, PDB: 6SSM), aligned to the NTD (left), CTD (middle), and NTD/CTD separately (right). **b**, **c** Arrangement of HERV-K immature (**b**) and mature (**c**) lattices. NTD is colored in blue/cyan, CTD in orange/red. **d**, **e** Detailed views of the dimer (top) and trimer (bottom) interface in the immature (**d**) and mature (**e**) lattices.

a "type-D" betaretrovirus, which is distinct from MMTV and other betaretroviruses. RSV, an alpharetrovirus, is placed as having the closest relationship to HERV-K. Indeed, it overlaps well in both domains and its trimeric interface is formed similarly as well (Fig. 5e). Even with these similarities, the quaternary assemblies differ between the two viruses, including the 6HB. MLV is situated at the other end of the tree, which can be explained by it neither overlapping well on its CTD nor NTD (Fig. 5a). MLV's distant relationship has also been characterized when comparing mature forms[18]. Interestingly, among all immature retroviruses, the dimeric interfaces are conserved, even comparing with the distant MLV (Fig. 5f).

Studying the structure of an endogenous virus provides insights into the architecture of endogenous viruses when they integrated into the genome. HERV-K is considered a young endogenous virus as some copies integrated <1 million years ago[38]. A comparison of the HERV-K CA structure to those from exogenous viruses shows how they are different. However, as the consensus sequence of HERV-K is used here, which is an approximation of the HERV-K group, small deviations might be possible, especially when considering that the endogenized viruses might have resulted from a selection of outliers more prone to endogenization. Nonetheless, the overall immature virus architecture is conserved. Intriguingly, a host small molecule is recruited to assist the assembly of immature HERV-K in the same way as the other retroviruses, suggesting a hallmark of immature retroviruses across genera and evolutionary time. The apparent structural conservation reinforces the note that the assembly and maturation processes among retroviruses involve similar mechanisms.

## Methods

### VLP production

The consensus HERV-K gag/pol gene *HERV-Gag-PR-Pol*[1] was kindly provided by Paul Bieniasz. Transfections were done in HEK293T cells (ECACC) with the consensus HERV-K Gag plasmid using GeneJet as a vessel. The VLPs were harvested 48 h post-transfection by spinning at low speed and filtering the supernatant to remove large cellular debris. The supernatant was then underlayed with 8% OptiPrep (Sigma-Aldrich) in STE buffer (100 mM NaCl, 10 mM Tris-Cl, pH 8.0, 1 mM EDTA; Sigma-Aldrich) as a cushion and spun at 100,000 g, 4 °C for 1:20 h. The pellet was spun onto 10%, 20% and 30% OptiPrep gradient at 120,000 g, 4 °C for 2:30 h. The VLP bands were extracted with a syringe and pelleted again in STE at 160,000 g, 4 °C for 1:20 h to remove residual OptiPrep. The pellet was then resuspended in residual liquid.

### CryoET sample preparation

VLPs were mixed with 6 nm gold fiducials. The plunge freezing was done with a Leica GP2 by blotting from the non-carbon side. The sample was applied onto lacey carbon grids, 300 mesh Cu (Agar Scientific), blotted for 3 s and plunged into liquid ethane. Grids were screened on a Talos Arctica.

### Data collection

Data collection was done on a Titan Krios equipped with a Selectris energy filter (Thermo Fisher Scientific). Pixel size was set to 1.5 Å, exposure at 1.17 s, a total dose of 127.5 electrons/Å² and a slit width of

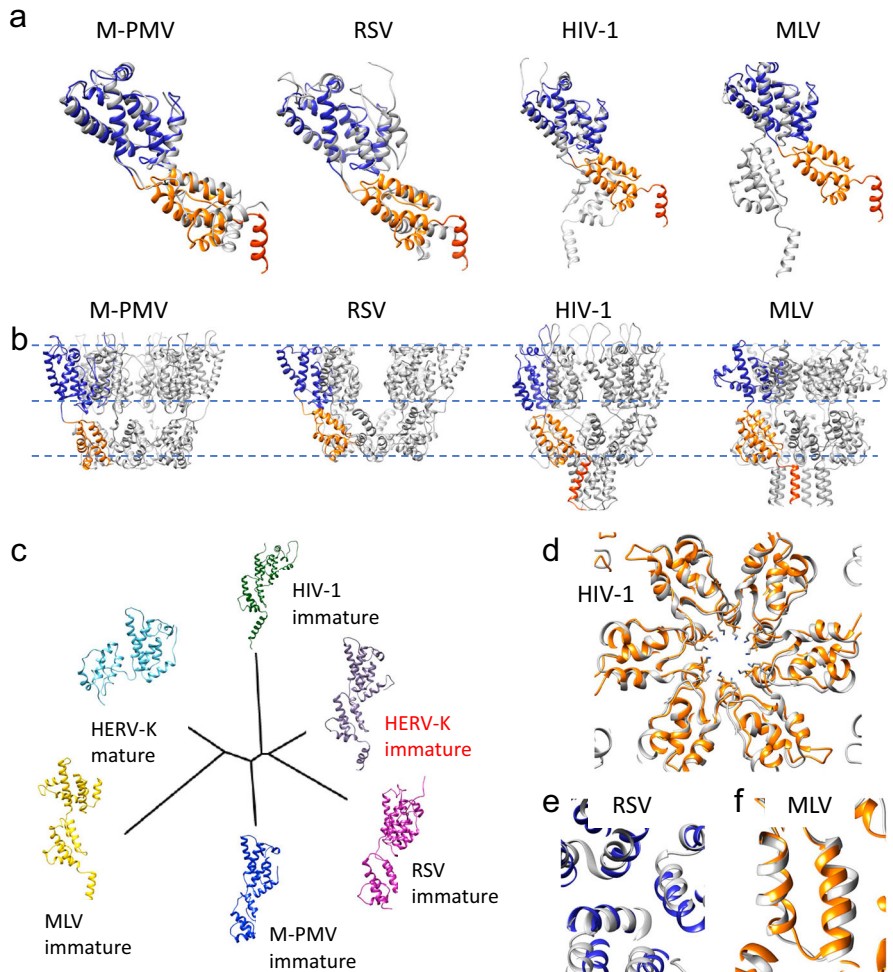

**Fig. 5 | Conservation among the immature retroviruses. a** Overlay of the immature CA monomer structures of HERV-K (NTD blue, CTD orange, h12 red) with those of other retroviruses (gray) aligned to the NTD, shown are M-PMV (PDB: 6HWI, beta-retrovirus), RSV (PDB: 5A9E, alpha-retrovirus), HIV-1 (PDB: 7ASL, lentivirus), MLV (PDB: 6HWW, gamma-retrovirus). **b** Hexamer structures of M-PMV, RSV, HIV-1 and MLV. **c** Structure based phylogenetic tree. **d–f** Conservation of immature CA intermolecular interfaces, compared with HIV-1 (hexamer interface, **d**), RSV (trimer interface, **e**) and MLV (dimer interface, **f**). HERV-K structure is colored in blue and orange, other retroviruses in gray.

5 eV. The tilt series were collected with a dose symmetric tilt scheme using a +/−60° range and a step size of 3°.

**Tomogram segmentation and subtomogram averaging**
124 tilt-series were aligned and reconstructed using automated routines implemented in the IMOD package[39]. Astigmatic per-tilt CTF models were estimated using geometric parameters obtained from the tilt-series alignment process[40]. To select the positions corresponding to Gag protein, VLPs were first manually picked from each tomogram in 3D. The radius of each VLP was estimated and saved together with their xyz-center coordinates. Volume segmentation along the low-density CA lattice was performed semi-automatically using minimal geodesic surfaces[41,42]. Uniformly distributed positions sampled along the CA lattice were generated while maintaining a minimum inter-unit distance of 96 Å. Using this strategy, 274,994 particle positions were selected from the segmentation of 794 VLPs and corresponding sub-volumes were extracted for further analysis. Ab-initio sub-volume averaging was done using EMAN2[43], followed by high-resolution refinement using constrained single-particle tomography (CSPT)[40,44]. After several rounds of particle orientation and tilt-geometry refinement, low-quality and duplicate particles were removed, resulting in 188,111 clean particles that were used to produce the final reconstruction of immature Gag at 3.2 Å resolution.

Resolution was estimated using the half-map Fourier Shell Correlation (FSC) using the 0.143-cutoff criteria. The final EM map and atomic coordinates were mapped back into the tomograms using the ArtiaX plugin[45] with the particle orientations obtained during CSPT refinement[42].

**Model building and refinement**
We built and refined a structure model of the HERV-K CA hexamer prior to MD simulations via comparative modeling using RosettaCM[32] according to the following procedure.

First, the cryoET STA structure of the HERV-K CA hexamer was rigid body fitted separately into the density using the UCSF Chimera[46] software version 1.17 by generating an initial coarse simulated density map at 8 Å resolution, fitting it into the experimental cryoET density and then sequentially increasing the resolution of the simulated density map to 4 Å and 2 Å and then fitting again. Furthermore, the experimental cryoET density within 2 Å of the HERV-K CA hexamer was selected and used for structure refinement. In addition, a fitted CA monomer was extracted from the fitted structure and used as a template for the density guided comparative modeling.

Next, the HERV-K CA amino acid sequence was threaded into the template monomer. Furthermore, six chains of the threaded model were fitted back into the hexamer cryoET density and used to generate

symmetry definitions for use in Rosetta, restraining adjacent chains to maintain a C6 NCS symmetry.

RosettaCM was then used to refine the model using the cryoET density by performing three steps of optimization: firstly, de novo structures for missing loops were generated and optimized in the torsional angle and cartesian coordinate space via Monte Carlo sampling. Secondly, the generated loops were joined to the structures sampled from the template model by performing Monte Carlo sampling in the cartesian coordinate space, optimizing the backbone geometry and backbone hydrogen-bonding interactions. In the final step, the side-chains of the full structure were optimized geometrically in the Cartesian coordinate space by minimizing the Rosetta all-atom energy function[47]. The three steps used the elec_dens_fast density scoring term with weights of 10, 10 and 50, respectively. In addition, symmetric scoring and optimization was enabled in all steps. The last all-atom density scoring function used the talos_2014 weights. The three optimization steps were handled on a single procedure using the hybridizer mover in Rosetta scripts release 2021.16.61620.

In total, we independently generated 800 models using this procedure and the top model with the lowest potential energy as measured by Rosetta's all-atom energy function[47] was selected as the initial structure for MD simulations.

The quality of the refined structure was quantified using the MolProbity[48] server, which evaluates the structural consistency of the bonds, angles, dihedrals and rotamers of a protein structure compared with their experimentally expected values[49,50]. Although the starting unrefined structure contained abundance of steric clashes with atoms at distances lower than 0.4 Å as well as outlier conformations for the bonds, angles, dihedrals and rotamers, the geometric refinement in both Cartesian and torsional angle space allows us to obtain a structure with minimal clashes, increasing the percentage of favored dihedrals and rotamers and reducing the number of structural outliers while conserving the overall secondary structure and being consistent with the cryoET density. As a result, our refined model achieves a lower MolProbity score as summarized in Supplementary Table 1 and is well fitted into the density as calculated by Q-scores[51] (Supplementary Fig. 12). The resulting model is presented in Supplementary Fig. 4.

## System setup

The atomic model of the HERV-K CA hexamer which was generated using the Rosetta software based on the cryoET density was used as the initial model for MD simulations (Supplementary Fig. 5a). The modeled CA hexamer was used to generate a hexamer of hexamers (HOH) motif (six hexamers around a central hexamer) by fitting each modeled hexamer into the cryoET density using the UCSF Chimera software[46] (Supplementary Fig. 5b). We docked the IP6 molecules between Lys166 and Lys240 rings in each hexamer using the Autodock Vina[52] tool implemented in the UCSF Chimera software. We placed a hundred chloride ions and fifty sodium ions near the hexamers using the Cionize plugin in the VMD software[53] version 1.9.4a57. Further, we solvated the resulting system with TIP3P water molecules[54] and added additional sodium and chloride ions to achieve a salt concentration of 150 mM using autoionize plugin in VMD[53] (Supplementary Fig. 5c). Thus, the prepared HOH model with solvent had all-atom resolution granularity necessary to quantify the stability of the structure at a residue level and to identify electrostatic, hydrogen bonding and hydrophobic interactions. The final simulation domain contained ≈1,250,000 atoms with the overall system size of ≈312 Å × 303 Å × 140 Å. A system without IP6 molecules was generated following the same setup protocol and had a similar system size as detailed in Supplementary Table 2.

## Simulation details

We subjected the resulting HOH simulation domains in the presence and absence of IP6 molecules to the same simulation protocol. At first, the simulations domains were subjected to minimization in two steps

using the NAMD2.14 simulation software[55]. In the first minimization step, we fixed the protein and the IP6 molecules while the solvent molecules were energy minimized using the conjugate gradient scheme until the gradient converged to values below 10 kcal mol-1 Å-1. In the second step, restraints on the protein and IP6 molecules were released and the entire system was energy minimized for 100,000 steps using the conjugate gradient scheme. Each minimization step was followed by thermalization of the simulation domain during which the system was heated from 50 K to 310 K in the increments of 5 K over 0.5 ns. During the second thermalization step, we applied the positional restraints to the heavy backbone atoms of the hexamers and to the heavy atoms of the IP6 molecules with a force constant of 10 kcal mol-1 Å-1. Subsequently, we conducted molecular dynamics simulations during which the positional restraints were gradually removed at a rate of 1 kcal mol-1 Å-1 per 0.2 ns from 10 kcal mol-1 Å-1 to 0 kcal mol-1 Å-1 over 2 ns. The systems were then equilibrated for 2 ns. We conducted the latter two simulations in the NPT ensemble while the temperature was maintained at 310 K with a coupling constant of 1 ps-1 using the Langevin thermostat and the pressure was maintained at 1 atm with period and decay parameters set to 100 ps and 50 ps, respectively, using the Nose-Hoover barostat. During the equilibration simulations, the gridForces[56,57] were enabled to maintain the overall HOH assembly and prevent diffusion of the individual monomers into the solvent using the experimental cryoET density.

After performing the equilibration of the simulation domains, we conducted production simulations of the HOH systems. We applied the positional restraints of 1 kcal mol-1 Å-1 to the Cα atoms in the protein backbone of the helical segments in each hexamer. Periodic boundary conditions were used in all simulations with a timestep of 2 fs. The electrostatics interactions were calculated using the particle mesh Ewald method with a cutoff for short-range electrostatics interactions set to 12 Å. We conducted all simulations using the CHARMM36m[58] force field for proteins, the force field parameters for IP6 were generated using CGenFF[59,60], the Roux force field parameters for ions[61], and the TIP3P water model[54]. We generated 200 ns worth of data for each of the HOH systems in the presence and absence of the IP6 molecules.

## Interactions between amino acids

We analyzed interactions formed between amino acids at the trimer and dimer interfaces established by the neighboring hexamers as well as the intradomain interactions. For residues at inter-hexamer dimer interfaces and inter-hexamer trimer interfaces, their root mean square fluctuation and contacts with other residues at the interface were collected over all 12 dimer interfaces and 6 trimer interfaces, respectively, while interaction distances and root mean squared fluctuations for intra-hexamer interface residues were collected from all 7 hexamer units in the HOH assembly, values reported are statistical averages over all copies of a residue in a given interface and are therefore averages over independent replicates with different initial coordinates. Hydrophobic interactions were captured by computing center-of-mass to center-of-mass distances between the heavy atoms of the interacting amino acids side chains while salt bridging interactions were defined as the distances between the interacting atoms.

## Western blot

Mutations were introduced by designing forward and backward primers which included the desired mutation and then doing a PCR using Pfr Turbo and restricting the original plasmid with DpnI at 37 °C. Primer sequences are provided in the Source Data File. The mutations were confirmed by sequencing (Forward primer: TCAAGAA AGGAAGGAGATACTGAG). Transfections were done as explained in the VLP preparation section but with 10× less cells at the start. The HERV-K_con plasmid was used as a positive control and no plasmid was used for the negative control. The supernatant was spun down at low

speed, filtered and pelleted onto an 8% OptiPrep cushion at 150,000 g for 1 h. The supernatant was run on a SDS gel and then transferred onto a membrane. The membrane was blocked with 5% skimmed milk and then incubated with the primary HERV-K Gag antibody (1:5000 dilution, HERM-1841-5) overnight at 4 °C. The membrane was washed previously to incubation with the secondary anti-mouse HRP antibody (1:5000 dilution, A0168 Sigma-Aldrich) for an hour at RT and imaged with the Clarify Western ECL substrate solutions (Bio-Rad). For an example of full scan blots, see the accompanying source data file.

### Phylogenetic analysis
The structure based phylogenetic tree was done using SHP[62,63], as specified in[64]. The following pdb files were used: 6SSM (HERV-K mature)[18], 7ASL (HIV-1 immature)[30], 6HWW (MLV immature)[36], 6HWI (M-PMV immature)[36], 5A9E (RSV immature)[37], this HERV-K immature model. If the pdb files contained more than one chain, only one monomer was used and the monomers were pre-aligned in Chimera to get them into the same coordinate system before running SHP. The amino acid sequence alignment was done using the Clustal Omega online tool and the amino acid coloring was done using MView[65].

### Reporting summary
Further information on research design is available in the Nature Portfolio Reporting Summary linked to this article.

## Data availability
All data needed to evaluate the conclusions in the paper are present in the paper and/or the Supplementary information and source data are provided with this paper. The cryoET subtomogram averaging density maps and corresponding atomic models have been deposited in the EMDB and PDB, with accession codes EMD-16511 and PDB ID 8C9M, respectively. PDB codes of previously published structures used in this study are 6SSM (HERV-K mature), 7ASL (HIV-1 immature), 6HWW (MLV immature), 6HWI (M-PMV immature) and 5A9E (RSV immature). Source data are provided with this paper.

## Code availability
All input files and scripts used for model refinement and molecular dynamics simulation as well as initial and final coordinates of the analyzed trajectory have been deposited in the public repository Zenodo [https://doi.org/10.5281/zenodo.8180645].

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

## Acknowledgements

We thank Paul Bieniasz for the pCRVI/Gag-PR-Pol plasmid, Geoff Sutton and Weng Ng for help with the structure-based phylogenetic tree plot. We thank Luiza Mendonca for help with initial VLP preparation. We acknowledge Diamond for access and support of the cryo-EM facilities at the U.K. national electron Bio-Imaging Centre (eBIC) (proposals NT21004 and NR21005), funded by the Wellcome Trust, MRC, and BBSRC. The data processing aspects of this research were supported by the Wellcome Trust Core Award Grant Number 203141/Z/16/Z and the NIHR Oxford BRC. We acknowledge computational support through the following resources: the Delaware Advanced Research Workforce and Innovation Network (DARWIN) and the Caviness cluster. This study utilized the computational resources offered by Duke Research Computing (http://rc.duke.edu). We thank Tracy Futhey, Katie Kilroy, Charley Kneifel, Mike Newton, Victor Orlikowski, Tom Milledge, and David Lane from the Duke Office of Information Technology and Research Computing for providing assistance with the computing environment. This research was funded by the UK Wellcome Trust Investigator Award (206422/Z/17/Z) and the ERC AdG grant (101021133), A.S.K. was supported by a Wellcome Trust Cellular Structural Biology Dphil Studentship. Research reported in this publication was partially supported by NIAID and NIGMS of the National Institutes of Health (R01AI157843 and R01GM141223). This work used Delta at the National Center for Supercomputer Applications through allocation MCB170096 from the Advanced Cyberinfrastructure Coordination Ecosystem: Services & Support (ACCESS) program, which is supported by National Science Foundation grants #2138259, #2138286, #2138307, #2137603, and #2138296.

## Author contributions

A.S.K. and P.Z. conceived project. A.S.K. produced and purified HERV-K VLPs, prepared the cryoET sample with help from J.S. A.H. collected the tomography tilt series data. H.F.L. and Y.Z. did the subtomogram averaging. J.S.R., L.L. and J.R.P. produced the molecular dynamics simulations. A.S.K. and P.Z. analyzed the HERV-K structure. A.S.K., L.L., J.S.R., J.R.P., A.B., and P.Z. wrote the paper with input from all authors.

## Competing interests

The authors declare no competing interests.
