## [Peer Review File · Nature Communications]

Molecular architecture and conservation of an immature human endogenous retrovirusREVIEWER COMMENTS

Reviewer #1 (Remarks to the Author):

In the manuscript by Krebs et al, they determine the structure of the human endogenous retrovirus K, using cryo-ET and MD simulations. This is an interesting system as the virus, at some point in the distant past, integrated into the human genome and VLP particles are expressed under certain disease conditions. Furthermore, increasing our general knowledge about retrovirus structure is important for understanding this family of viruses which have had severe impacts on global human health. The paper is fairly straight forward, I am impressed by the resolution that is achieved using the cryo-ET method, as the paper mainly focuses structural details and stabilizing interactions. Overall, I don't have any major concerns about the quality of the work, but there are two points that I think have a moderate degree of significance that should be addressed

1) The details of the western blot are not described and I find the results somewhat baffling (this maybe my naivete as I don't have direct experience here). Why would mutations that do affect VLP formation form dimer and monomer bands, while those that inhibit VLP formation do not show either a monomer or dimer band? It would think if VLPs are not forming then monomers must still be detectable?

2) Methodologically , I find the choice to positionally restrain the C α atoms to be a very restrictive choice that will severely limit the dynamics of the system. Can the authors provide a rationale/justification for making such a choice, rather than letting the system evolve natively? These restraints will certainly have an impact on the RMSF, RMSD and interactions distances.

Minor points:

The abstract contains many abbreviations without definition (e.g. CA, MA, SP1, p15, IP6), I'm not sure if this is appropriate or not.

Line 45 – Should be "rendering"

Line 70 – What is the meaning of the subscript "con" in HERV-K_{con} ?

Line 83-85. In this last sentence are you say the IP6 and 6HB are serving the same role in different systems or does HERV-K also have IP6 ? The sentence is not written clearly to understand the point here.

Supp Fig 2. There is not "b" label in the figure.

Results - As I read through the first two section of the results I have some confusion. In Figure 2 and Figure S2, it is unclear to me if this is the inner core (CA) or the MA or is it the full construct because it has not been cleaved yet? I find it strange that the terms CA or MA are not used in describing the protein structure in the captions, but maybe that is because it is not cleaved and the term Gag is appropriate. And then in Figure S3 it is showing the mapping back from the STA to the full tomogram, so in the particles shown in Figure 2A are these icosahedrally(or dodecahedrally) averaged, but not truly icosahedral in the structure? Some of these questions maybe obvious to a retrovirus expert, but for a more general audience I think this should be clarified.

Line 177 – Y66 is listed in a list of charged residues. I presume it is not charged.

Figure 3G –Positive and negative controls should be explained. For each of the mutations it should be listed as Y66A for example if they are all mutations to Alanine.

Reviewer #2 (Remarks to the Author):

Krebs et al

Krebs and colleagues describe a structure for the immature capsid assembly of HERV-Kcon, a previously described synthetic consensus of multiple human HERV-K(HML2) loci. Although there are caveats to drawing general conclusions about all HERV-K(HML2) elements or the viruses that produced them from one consensus, the structure is a breakthrough in allowing a comparison to mature forms of capsid and to the mature and immature structures of other retroviruses. Certainly many of the conserved features suggest that the structure is reasonable, and it allows for hypotheses about assembly and maturation, particularly of betaretroviruses. The structural data are solid, and likely will be of great interest to the field.

My primary concern is not with the data or the structural conclusions, but with multiple points in the abstract and discussion that seem to imply that HERV-K represents an ancestral form of "modern" retroviruses - it does not. For all intents and purposes, HERV-K(HML2) is a sister taxon to extant exogenous retroviruses, not a predecessor. To the extent that it has unique properties that distinguish it from other retroviruses, these distinctions likely existed a million years ago, too. This is apparent when including HERV-K in phylogenetic trees, where it represents a terminal branch just like MLV, HIV, MPMV, MMTV, etc. The differences between these viruses likely go back hundreds of millions of years, and are not the result of recent evolution (in the last million years). The real value in the paper is that this represents another, unique structure that can contribute to comparative analysis of retroviruses, and in particular, it provides a structure for a unique form of betaretrovirus that assembles at the plasma membrane.

Specific comments:

Lines 24-25: I caution the authors against clear statements associating HML-2 with specific disease states and aging; most of these studies remain correlative and controversial.

Line 25, lines 29-30: the contrast between ancient and modern is likely to be misleading - differences are more likely to reflect the same kinds of differences that one would find when comparing modern viruses. I'd be very surprised if the differences have anything to do with being "ancient" versus "modern".

Line 37 - in place of "modern" might try "other" or "exogenous". Modern is a false distinction here, suggesting that HERV-K and extant retroviruses represent two different stages on an evolutionary tree.

Line 46 - "HERV-K" and "HML2" are not interchangeable, so "aka" is inappropriate here. HERV-K includes maybe a dozen families, of which HML-2 is one.

Lines 48-49: The statement is confusing. HERV-K is a family of tens of thousands of elements, none of which contain all the viral ORFs intact. Referring to "HERV-K" as being expressed makes it sound like the entire family is expressed and/or that the authors are dealing with a single HERV-K locus. The authors should be careful to use language that distinguishes between properties of the HERV-K family, the HERV-K(HML-2) group, or individual HERV-K(HML2) loci. In fact, this is why the paper is based on a consensus HERV-K(HML2) genome known as HERV-Kcon.

Lines 56-60 - These lines are actually ok. The use of "ancestor" and "fossils" here is appropriate,

and the wording is fine. It is true that HERV-K(HML2) elements can be thought of as ancestors or fossils of a specific viral lineage (although we do not know of any living descendants at this time).

Lines 146-148: Wording again seems to imply that HERV-K represents an ancestral form of modern betaretroviruses like MPMV, when in fact the two (HERV-K and MPMV) are both evolved significantly from a common ancestor. It is entirely plausible that MPMV "lost" the 6HB long before HERV-K endogenized in humans. Given how few betaretrovirus structures there are, we also don't currently know which represents the most common form - with a 6HB or without a 6HB. It is risky to assume for example that MPMV and MMTV represent "normal" betaretroviruses, due to sampling bias (we only study those viruses that have been identified; we don't know if these are representative of the betaretroviruses as a whole).

Line 162 - "indicative" should maybe be "indicator"?

Line 169 - "western blot" is not a proper noun and shouldn't be capitalized (although this is likely up to the editorial staff of the journal).

Line 239 - "modern". very risky to think of HERV-K as "ancient" versus exogenous viruses as "modern". It is very likely that "modern" retroviruses have the same structures now that they did tens or hundreds of millions of years ago. HERV-K is just different, not an evolutionary stage leading to modern retroviruses. In fact, it is even possible that exogenous HERV-K still exists in nature.

Line 251 - it might be worth mentioning that MPMV is a "type-D" betaretrovirus, which is distinct from MMTV and other betaretroviruses in multiple ways.

Lines 265-266: This line in the discussion gets right to the core of my concern. It is basically incorrect. This would only be true if HERV-K was an ancestor of the viruses it is being compared to, which is very unlikely. In a betaretrovirus phylogenetic tree, HERV-K is not near the base (as an ancestor would be), rather it is the tip of a terminal branch, a relative of MMTV, MPMV, etc. An analogy would be the relationship between humans, neanderthals and denisovans - just because the latter two are extinct does not mean they were human ancestors. In fact, they co-existed with modern humans and on a phylogenetic tree are as "evolved" as humans. This is very likely the case when comparing HERV-K to extant betaretroviruses, such as MMTV and MPMV.

Discussion - Two critical caveats need to be discussed - 1) the possibility that HERV-Kcon might not be representative of HERV-K(HML2) viruses, because it is a consensus and therefore there is always the possibility that bias in the input or construction of the consensus means it is an outlier in some feature(s). 2) the possibility that endogenization resulted in selection of outliers that differed in some way from their exogenous forms, or that post-endogenization evolution altered the HERV-K sequences such that they might differ from their viral origins.

The paper is solid and the structure is very useful, but these caveats need to be acknowledged to avoid misunderstanding by readership, and the authors are encouraged to avoid language that implies HERV-K(HML2) is an evolutionary stage distinct from modern viruses.

Methods - The cited references do not give any useful information about SHP, making it difficult to judge the methodology or its application. At best, these papers simply refer to SHP in a single sentence, with no details about the program or the basis upon which it works. A link to the program or a paper that describes the program would be useful, or barring that, some description of the program should be included in the methods.

Reviewer #3 (Remarks to the Author):

This manuscript describes a cutting-edge methodology that uses cryoET and subtomogram averaging to obtain a high-resolution 3-D density map (3.2Å) of the CA protein of an immature human endogenous retrovirus. The authors then used this map as a constraint to build an atomic model for the protein, which was subjected to molecular dynamics simulations to explore the

flexibility of the side chains located in the subunit interface. Some of these side chains implicated to involve hydrophobic or hydrophilic interactions were experimentally mutated with alanine to evaluate if these interactions are necessary to capsid formation. These studies provide new insights into the critical amino acids involved in CA capsid formation. The subsequent comparison of this structure with those from other retroviruses offers a glimpse into the evolutionary relationship of the CA protein before its incorporation into the human genome. The results of this analysis have potential implications for the development of novel therapies for retrovirus and possibly other human diseases.

To improve readability and clarity of the submitted manuscript, I suggest the authors consider the following edits in their revised manuscript. Avoid using jargon and acronyms without proper explanation to make them more accessible to a wider audience, who may not be familiar with the retrovirus structure. Ensure that the flow of the manuscript is logical and easy to follow for the general readership.

The followings are my specific technical comments:

1. Line 111: The statement "The observed large distance between the membrane and the CA lattice in HERV-K corresponds well with the presence of these additional domains between MA and CA" would be more convincing if experimental data can be generated to knock out SP1 and p15 proteins and observe if the gap narrows. Additionally, can you detect any density corresponding to these domains of gag in your current map?
2. Line 117: Can you provide more information on how the 270K subvolumes were segmented and selected on the surface of 800 VLPD? How large is each subvolume? Is the targeted subvolume feature visible to the human eye, or was segmentation carried out in an iterative process using software? Although the supplementary methods provide references to answer these questions, a couple of sentences describing the methodology in the main text would be helpful to general readers.
3. Line 119: What does "high resolution constrained alignment of 2D projections" mean?
4. Line 122 "refined subtomogram positions are arranged in a hexagonal lattice with incorporated holes to accommodate the lattice curvature" - Does this mean there are really holes in the density (or low density) in those locations, or could it be caused by particles not aligned properly there? Some particles seem to be overlapping in Supp. Fig. 2c. Is this caused by the alignment error, or the STA does not represent the actual density well in those locations?
5. Line 126: Please state the molecular weights or the number of residues of the CA domain of Gag and explain why only the CA domain of Gag is resolved? Since only the CA structure is resolved, it would be more accurate to state throughout the text that the CA, but not the gag structure of the immature retrovirus, is solved in this study. Additionally, could you comment on the prospect of retrieving the 3D structures of the remaining components of the gag polyprotein diagrammed in Fig. 1a?
6. Line 136: Please define IP6 in terms of its chemical identity and size here rather than shown in Fig. 3f cited later in line 198 without any comment of this molecule. How do you know the putative density is an IP6 molecule? Do you have chemical evidence rather than inference from other structural/chemical studies? If the density map is not clear on the distance between the IP6 and the interacting residue side chains, it appears highly speculative in this map resolution to assert it to be IP6 and assign its chemical role. Perhaps the proposed IP6 molecule shown in Fig. 3f could be shown together with the density in Fig. 2f, so it would be clearer how the molecule fits the density. The molecular model shown in Fig. 3f does not look 6-fold symmetric like the density; perhaps only parts of it are resolved, e.g. the carbon ring, whereas the oxygen groups are not resolved because they may be more dynamic; this would be useful to clarify.
7. Supplement Table 1 shows the quality of the model, but some data or statement to demonstrate that the model fits the map is necessary, for example per-residue Q-scores. All cryoEM structures deposited to PDB undergo such measures (e.g., see the end of their submitted

PDB report, where it is colored on the backbone). To make it clearer, it could be plotted per residue (see the paper by S. Burley et al. Biophysical Reviews 2022), to indicate which residues and which parts of the model are resolved as expected and well-fitted to the map.

8. The FSC curve shown in Supp. Fig. 2b (the label b is missing) seems to dip at resolution approaching the reported resolution of 3.2 Å - could this be an effect of STA processing?

9. The map-model FSC shown in Supp. Fig. 2c indicates a resolution closer to 3.8 Å. Why is the 0.3 threshold used in Fig. 2c? Typically a threshold of 0.5 is used for map-model FSC; a reference if available for the use of 0.3 would be of general interest.

10. Line 157: The cryoEM map represents the mass density (often abbreviated as density) or potential function of the object, not the electron density of the object itself (see the paper by Unwin and Henderson JMB 1975 or the textbook by Glaeser, et al.).

11. Line 171: it states that "The chemical nature of these residues is not essential for trimer formation". I think it is more appropriate to describe it as a trimeric interface rather than trimer formation, which leaves an implication that trimer formation is the initial/critical step of the assembly. Is there a previous study suggesting whether the inter-dimer within a hexamer or the inter-trimeric among three hexamers is the driving force to initiate the entire virion assembly? It would be interesting to include some discussion on the assembly mechanism of retrovirus which might already be discussed in the literature?

12. Line 160: The simulations are targeted to characterize the monomer interface either within a hexamer or between hexamers. The authors should clarify in their description which interface is referred to as the intra or inter-protein interface, particularly for the dimer interface.

13. Line 163-171: Q118 and R100 in the trimer interface are found to be flexible as measured by RMSF in their simulation and non-essential in the particle assembly in the mutant experiments. This is a well-designed study. Are the densities of these residues less resolved in the map compared to those of other amino acids that are found to be more essential and less flexible?

14. Line 192: How was it confirmed that a sodium ion is involved in the interaction between the E132 of neighboring hexamers? Can you detect Na⁺ in the map?

15. Could you explain in the main text how the plots shown in Suppl. Fig. 6 were generated and how the distances vs flexibility is interpreted ?

16. Line 209: h12 is not annotated in Fig. 4a or the figure caption.

17. Line 217: Is there another way to describe this packing as domain-swapping (i.e., NTD-1 interacts with CTD-2 from proteins 1 and 2)? It may be easier to see this in Fig. 4 if there was a panel where each protein was annotated in a different color.

Point-to-point responses to the reviewers' comments

Reviewer #1 (Remarks to the Author):

In the manuscript by Krebs et al, they determine the structure of the human endogenous retrovirus K, using cryo-ET and MD simulations. This is an interesting system as the virus, at some point in the distant past, integrated into the human genome and VLP particles are expressed under certain disease conditions. Furthermore, increasing our general knowledge about retrovirus structure is important for understanding this family of viruses which have had severe impacts on global human health. The paper is fairly straight forward, I am impressed by the resolution that is achieved using the cryo-ET method, as the paper mainly focuses structural details and stabilizing interactions. Overall, I don't have any major concerns about the quality of the work, but there are two points that I think have a moderate degree of significance that should be addressed

1) The details of the western blot are not described and I find the results somewhat baffling (this maybe my naivete as I don't have direct experience here). Why would mutations that do affect VLP formation form dimer and monomer bands, while those that inhibit VLP formation do not show either a monomer or dimer band? It would think if VLPs are not forming then monomers must still be detectable?

The western blot is done not on the cells themselves, but on the concentrated media which would contain VLPs if they have been assembled and released. Thus, if VLPs cannot be assembled and released due to the mutations in the critical assembly interfaces, we cannot detect HERV-K Gag, dimer or monomer, in the media.

2) Methodologically, I find the choice to positionally restrain the C α atoms to be a very restrictive choice that will severely limit the dynamics of the system. Can the authors provide a rationale/justification for making such a choice, rather than letting the system evolve natively? These restraints will certainly have an impact on the RMSF, RMSD and interaction distances.

We thank the reviewer for their meaningful concerns. We utilized low energy 1kcal/mol/A restraints to the C α atoms in the protein backbone of the helical segments as a mean to maintain the structural information derived experimentally throughout the simulation, as the positions of helical structures were well characterized in the cryo-EM density. Adding restraints on the C α atoms allow us to effectively reduce the phase space and sample dynamics of the interfacial residues using the cryo-EM structure as reference. Furthermore, as the harmonic restraints were only applied on the C α atoms, they did not affect the RMSF or interaction distances measured for the residue sidechains, as these were measured from the positions of sidechain heavy atoms. Instead, we find low RMSF values to be related with stabilizing interactions between interface residues and with IP6.

Minor points:

The abstract contains many abbreviations without definition (e.g. CA, MA, SP1, p15, IP6), I'm not sure if this is appropriate or not.

We added definitions to these abbreviations.

Line 45 – Should be “rendering”

Corrected.

Line 70 – What is the meaning of the subscript “con” in HERV-K_{con} ?

It indicates the consensus sequence. We have rephrased the sentence that introduces the HERV-K_{con} for clarity in the revised manuscript.

Line 83-85. In this last sentence are you say the IP6 and 6HB are serving the same role in different systems or does HERV-K also have IP6 ? The sentence is not written clearly to understand the point here.

We changed that sentence to improve clarity.

Supp Fig 2. There is not “b” label in the figure.

Corrected.

Results - As I read through the first two section of the results I have some confusion. In Figure 2 and Figure S2, it is unclear to me if this is the inner core (CA) or the MA or is it the full construct because it has not been cleaved yet? I find it strange that the terms CA or MA are not used in describing the protein structure in the captions, but maybe that is because it is not cleaved and the term Gag is appropriate. And then in Figure S3 it is showing the mapping back from the STA to the full tomogram, so in the particles shown in Figure 2A are these icosahedrally(or dodecahedrally) averaged, but not truly icosahedral in the structure? Some of these questions maybe obvious to a retrovirus expert, but for a more general audience I think this should be clarified.

This is a complete Gag construct, but we just resolved the CA of Gag. The construct also contains uncleaved MA and NC, which are not resolved. We rephrased some parts to make it clearer.

Line 177 – Y66 is listed in a list of charged residues. I presume it is not charged.

Y66 is indeed not a charged amino acid, thanks for catching this. We have changed the text to reflect the hydrophobic nature of Y66.

“Charged residues R97, E69, Y66, K85 and D90 as well as hydrophobic residue Y66 were identified as not being flexible according to their low RMSF values (Fig. 3b).”

Figure 3G –Positive and negative controls should be explained. For each of the mutations it should be listed as Y66A for example if they are all mutations to Alanine.

We changed the labels to include A in the name and renamed the positive control. We also now specify in the Methods section what we used for positive and negative controls.

Reviewer #2 (Remarks to the Author):

Krebs et al

Krebs and colleagues describe a structure for the immature capsid assembly of HERV-Kcon, a previously described synthetic consensus of multiple human HERV-K(HML2) loci. Although there are caveats to drawing general conclusions about all HERV-K(HML2) elements or the viruses that produced them from one consensus, the structure is a breakthrough in allowing a comparison to mature forms of capsid and to the mature and immature structures of other retroviruses. Certainly many of the conserved features suggest that the structure is reasonable, and it allows for hypotheses about assembly and maturation, particularly of betaretroviruses. The structural data are solid, and likely will be of great interest to the field.

My primary concern is not with the data or the structural conclusions, but with multiple points in the abstract and discussion that seem to imply that HERV-K represents an ancestral form of "modern" retroviruses - it does not. For all intents and purposes, HERV-K(HML2) is a sister taxon to extant exogenous retroviruses, not a predecessor. To the extent that it has unique properties that distinguish it from other retroviruses, these distinctions likely existed a million years ago, too. This is apparent when including HERV-K in phylogenetic trees, where it represents a terminal branch just like MLV, HIV, MPMV, MMTV, etc. The differences between these viruses likely go back hundreds of millions of years, and are not the result of recent evolution (in the last million years). The real value in the paper is that this represents another, unique structure that can contribute to comparative analysis of retroviruses, and in particular, it provides a structure for a unique form of betaretrovirus that assembles at the plasma membrane.

We thank the reviewer for this comment. This is a very good point. We have rephrased the parts which were drawing any conclusion on an ancient retrovirus.

Specific comments:

Lines 24-25: I caution the authors against clear statements associating HML-2 with specific disease states and aging; most of these studies remain correlative and controversial.

We have revised accordingly.

Line 25, lines 29-30: the contrast between ancient and modern is likely to be misleading - differences are more likely to reflect the same kinds of differences that one would find when comparing modern viruses. I'd be very surprised if the differences have anything to do with being "ancient" versus "modern".

We have reworded these.

Line 37 - in place of "modern" might try "other" or "exogenous". Modern is a false distinction here, suggesting that HERV-K and extant retroviruses represent two different stages on an evolutionary tree.

We thank the reviewer and have reworded "modern" to "exogenous".

Line 46 - "HERV-K" and "HML2" are not interchangeable, so "aka" is inappropriate here. HERV-K includes maybe a dozen families, of which HML-2 is one.

We appreciate the reviewer's comment and have changed it accordingly.

Lines 48-49: The statement is confusing. HERV-K is a family of tens of thousands of elements, none of which contain all the viral ORFs intact. Referring to "HERV-K" as being expressed makes it sound like the entire family is expressed and/or that the authors are dealing with a single HERV-K locus. The authors should be careful to use language that distinguishes between properties of the HERV-K family, the HERV-K(HML-2) group, or individual HERV-K(HML2) loci. In fact, this is why the paper is based on a consensus HERV-K(HML2) genome known as HERV-Kcon.

We appreciate this comment. The complete paragraph has been rephrased to make the distinction clear.

Lines 56-60 - These lines are actually ok. The use of "ancestor" and "fossils" here is appropriate, and the wording is fine. It is true that HERV-K(HML2) elements can be thought of as ancestors or fossils of a specific viral lineage (although we do not know of any living descendants at this time).

Lines 146-148: Wording again seems to imply that HERV-K represents an ancestral form of modern betaretroviruses like MPMV, when in fact the two (HERV-K and MPMV) are both evolved significantly from a common ancestor. It is entirely plausible that MPMV "lost" the 6HB long before HERV-K endogenized in humans. Given how few betaretrovirus structures there are, we also don't currently know which represents the most common form - with a 6HB or without a 6HB. It is risky to assume for example that MPMV and MMTV represent "normal" betaretroviruses, due to sampling bias (we only study those viruses that have been identified; we don't know if these are representative of the betaretroviruses as a whole).

We appreciate the reviewer's comments and have reworded to avoid assuming the time frame when the 6HB was lost.

Line 162 - "indicative" should maybe be "indicator"?

Yes, replaced.

Line 169 - "western blot" is not a proper noun and shouldn't be capitalized (although this is likely up to the editorial staff of the journal).

Corrected.

Line 239 - "modern". very risky to think of HERV-K as "ancient" versus exogenous viruses as "modern". It is very likely that "modern" retroviruses have the same structures now that they did tens or hundreds of millions of years ago. HERV-K is just different, not an evolutionary stage leading to modern retroviruses. In fact, it is even possible that exogenous HERV-K still exists in nature.

We appreciate the reviewer's comments and have rephrased the term to address the issue of "ancient" versus "modern".

line 251 - it might be worth mentioning that MPMV is a "type-D" betaretrovirus, which is distinct from MMTV and other betaretroviruses in multiple ways.

We have added this distinction to M-PMV.

Lines 265-266: This line in the discussion gets right to the core of my concern. It is basically incorrect. This would only be true if HERV-K was an ancestor of the viruses it is being compared to, which is very unlikely. In a betaretrovirus phylogenetic tree, HERV-K is not near the base (as an ancestor would be), rather it is the tip of a terminal branch, a relative of MMTV, MPMV, etc. An analogy would be the relationship between humans, neanderthals and denisovans - just because the latter two are extinct does not mean they were human ancestors. In fact, they co-existed with modern humans and on a phylogenetic tree are as “evolved” as humans. This is very likely the case when comparing HERV-K to extant betaretroviruses, such as MMTV and MPMV.

We appreciate the reviewer’s comments and have rephrased that paragraph.

Discussion - Two critical caveats need to be discussed - 1) the possibility that HERV-Kcon might not be representative of HERV-K(HML2) viruses, because it is a consensus and therefore there is always the possibility that bias in the input or construction of the consensus means it is an outlier in some feature(s). 2) the possibility that endogenization resulted in selection of outliers that differed in some way from their exogenous forms, or that post-endogenization evolution altered the HERV-K sequences such that they might differ from their viral origins.

The paper is solid and the structure is very useful, but these caveats need to be acknowledged to avoid misunderstanding by readership, and the authors are encouraged to avoid language that implies HERV-K(HML2) is an evolutionary stage distinct from modern viruses.

We appreciate the reviewer’s input. We paid specific attention to these in the revision.

Methods - The cited references do not give any useful information about SHP, making it difficult to judge the methodology or its application. At best, these papers simply refer to SHP in a single sentence, with no details about the program or the basis upon which it works. A link to the program or a paper that describes the program would be useful, or barring that, some description of the program should be included in the methods.

We included an additional reference (Ng et al. 2020), which describes how structure based phylogenetic trees are generated.

Reviewer #3 (Remarks to the Author):

This manuscript describes a cutting-edge methodology that uses cryoET and subtomogram averaging to obtain a high-resolution 3-D density map (3.2Å) of the CA protein of an immature human endogenous retrovirus. The authors then used this map as a constraint to build an atomic model for the protein, which was subjected to molecular dynamics simulations to explore the flexibility of the side chains located in the subunit interface. Some of these side chains implicated to involve hydrophobic or hydrophilic interactions were experimentally mutated with alanine to evaluate if these interactions are necessary to capsid formation. These studies provide new insights into the critical amino acids involved in CA capsid formation. The subsequent comparison of this structure with those from other retroviruses offers a glimpse into the evolutionary relationship of the CA protein before its incorporation into the human genome. The results of this analysis have potential implications for the development of novel therapies for retrovirus and possibly other human

diseases.

To improve readability and clarity of the submitted manuscript, I suggest the authors consider the following edits in their revised manuscript. Avoid using jargon and acronyms without proper explanation to make them more accessible to a wider audience, who may not be familiar with the retrovirus structure. Ensure that the flow of the manuscript is logical and easy to follow for the general readership.

The followings are my specific technical comments:

1. Line 111: The statement "The observed large distance between the membrane and the CA lattice in HERV-K corresponds well with the presence of these additional domains between MA and CA" would be more convincing if experimental data can be generated to knock out SP1 and p15 proteins and observe if the gap narrows. Additionally, can you detect any density corresponding to these domains of gag in your current map?

We compared the segment between MA and CA among the known retroviruses and found that HERV-K has the largest segment, encompassing P1 and SP15. We agree that knocking out SP1 and p15 proteins would have been a definitive test, but this would be beyond the scope of this study. To be more precise, we have rephrased the sentence using "correlates", instead of "corresponds", in the revised manuscript.

2. Line 117: Can you provide more information on how the 270K subvolumes were segmented and selected on the surface of 800 VLPD? How large is each subvolume? Is the targeted subvolume feature visible to the human eye, or was segmentation carried out in an iterative process using software? Although the supplementary methods provide references to answer these questions, a couple of sentences describing the methodology in the main text would be helpful to general readers.

We appreciate the reviewer's comments and have added the following two sentences to the main text to address this point:

"The centroids of ~800 VLPs were computationally detected from the tomograms in an automated manner and used as seeds to find the location of the external surface of the CA lattice using unsupervised segmentation routines. 270K uniformly distributed positions were identified along the CA lattice of all VLPs and sub-volumes with a side length of 40 nm were extracted for further processing (Supplementary Fig. 2a)."

3. Line 119: What does "high resolution constrained alignment of 2D projections" mean?

We used this term to describe the approach we used for high-resolution STA, called constrained single-particle tomography (CSPT), originally described in [Bartesaghi et al., 2012] and recently extended in [Bouvette et al., 2021]. Unlike traditional sub-volume averaging, this approach uses 2D particle projections extracted directly from the raw tilt-series and enforces the geometry constraints during refinement to improve map resolution. We revised the text to clarify this point:

"Subtomogram averaging followed by analysis of the raw tilt-series projections using STA refinement procedures that impose the constraints of the tilt-geometry, resulted in a 3.2 Å resolution map (Fig. 2a, Supplementary Fig. 2b)."

4. Line 122 “refined subtomogram positions are arranged in a hexagonal lattice with incorporated holes to accommodate the lattice curvature” - Does this mean there are really holes in the density (or low density) in those locations, or could it be caused by particles not aligned properly there? Some particles seem to be overlapping in Supp. Fig. 2c. Is this caused by the alignment error, or the STA does not represent the actual density well in those locations?

During 3D refinement, not all sub-volumes extracted from the CA lattice were used for the final reconstruction. For example, particles with lower alignment scores or those that were too close to neighbouring hexamers were removed from the analysis. These particles were not mapped back into the VLPs, thus causing some of the “holes”. We changed “holes” by “gaps” to clarify this point. Each hexagonal model was coloured according to the alignment score, with red indicating lower correlation against the final map. The overlapping particles could be the result of inaccurate orientation assignment (orange and red particles) or leftover duplicates that were not successfully removed during the particle cleaning procedure. The map that was mapped back into the VLPs extended beyond the size of a single hexamer, also contributing to the overlap seen in Supp. Figure 2c.

5. Line 126: Please state the molecular weights or the number of residues of the CA domain of Gag and explain why only the CA domain of Gag is resolved? Since only the CA structure is resolved, it would be more accurate to state throughout the text that the CA, but not the gag structure of the immature retrovirus, is solved in this study. Additionally, could you comment on the prospect of retrieving the 3D structures of the remaining components of the gag polyprotein diagrammed in Fig. 1a?

We added the number of residues of the CA domain of Gag. We revised the wording throughout the manuscript to make it clear that only the CA structure is solved.

It will be very challenging to retrieve the 3D structures of the remaining components of the Gag polyproteins, based on our prior experience and current state of the art. However, we have recently resolved the HIV-1 MA structure to $\sim 8 \text{ \AA}$ resolution (as Briggs group). We plan to apply the same strategy to determine the structure of HERV-K MA. The NC part is highly flexible (based on HIV-1 NC) and has low molecular weight. It will be very difficult to resolve its structure by cryoEM, but perhaps other methods, such as NMR could work.

6. Line 136: Please define IP6 in terms of its chemical identity and size here rather than shown in Fig. 3f cited later in line 198 without any comment of this molecule. How do you know the putative density is an IP6 molecule? Do you have chemical evidence rather than inference from other structural/chemical studies? If the density map is not clear on the distance between the IP6 and the interacting residue side chains, it appears highly speculative in this map resolution to assert it to be IP6 and assign its chemical role. Perhaps the proposed IP6 molecule shown in Fig. 3f could be shown together with the density in Fig. 2f, so it would be clearer how the molecule fits the density. The molecular model shown in Fig. 3f does not look 6-fold symmetric like the density; perhaps only parts of it are resolved, e.g. the carbon ring, whereas the oxygen groups are not resolved because they may be more dynamic; this would be useful to clarify.

We thank the reviewer for their suggestions. We have modified the text to clarify IP6's chemical nature and molecular weight: “At the top of the 6HB, there is a strong density, most likely

corresponding to the 647.94 Da myo-inositol hexakisphosphate (IP6) molecule, coordinated by two rings of lysine residues, K166 and K240”.

Indeed, the map resolution is not enough to assign the density to IP6, but we attribute it most probably to IP6 based on several lines of additional supporting evidence: 1) extensive studies of HIV-1 immature Gag with IP6 (cryoET STA and crystal structures), 2) both HIV-1 Gag and HERV-K Gag have two rings of Lysine residues at the similar locations and a similar central density at the top of 6HB; 3) both HIV-1 and HERV-K Gag are produced the same way from HEK 293 cells; and 4) our all-atom MD simulations. The 3.2 Å resolution map does not resolve the atomic details of IP6 and the cryoEM map is six-fold symmetrized. Therefore, there are some differences between the cryoEM density and the non-symmetric atomic model of IP6 used in the MD simulations.

We agree with the reviewer and have overlapped the IP6 molecule with the density in Fig. 2e&f (see below).

The IP6 structure used in the MD simulations (shown in Fig 3f) does have all the six phosphoryl groups present, however, some phosphoryl groups are off plane, as is expected in myo-inositol hexakisphosphate.

While the aromatic ring of IP6 remains stable throughout the MD simulations, the oxygens in the phosphoryl groups are more flexible, consistent with the observed density. We added the new Supplemental Fig. 10 to illustrate the flexibility of IP6 atoms throughout the simulation.

7. Supplement Table 1 shows the quality of the model, but some data or statement to demonstrate that the model fits the map is necessary, for example per-residue Q-scores. All cryoEM structures deposited to PDB undergo such measures (e.g., see the end of their submitted PDB report, where it is colored on the backbone). To make it clearer, it could be plotted per residue (see the paper by S. Burley et al. Biophysical Reviews 2022), to indicate which residues and which parts of the model are resolved as expected and well-fitted to the map.

We thank the reviewer for this observation, we have followed the indicated reference and calculated the per-residue Q-scores for the model presented. We calculated an average Q-score of 0.461, higher than the 0.410 expected for a structure of 4.0 Å resolution, in particular, we observe good fitting for helical regions and lower Q-scores for flexible linkers. We included this information as a new Supplementary Fig. 12.

8. The FSC curve shown in Supp. Fig. 2b (the label b is missing) seems to dip at resolution approaching the reported resolution of 3.2 Å - could this be an effect of STA processing?

We added the missing label to this Figure panel. There could be several factors that contributed to the dip in the FSC, including the resolution-lowering effect of conformational heterogeneity present in the native HERV-K Gag complex, the inclusion of lower quality or misaligned particles during the reconstruction stage, and the fact that the reported resolution is already close to the Nyquist limit. Despite many efforts to improve map resolution by including higher resolution information during refinement, our STA processing routines could not recover information beyond 3.2 Å.

9. The map-model FSC shown in Supp. Fig. 2c indicates a resolution closer to 3.8 Å. Why is the 0.3 threshold used in Fig. 2c? Typically a threshold of 0.5 is used for map-model FSC; a reference if available for the use of 0.3 would be of general interest.

Although several publications used 0.3 cutoff for map-model FSC, we agree with the reviewer to use a threshold of 0.5, which corresponds to a resolution of 4.0 Å.

10. Line 157: The cryoEM map represents the mass density (often abbreviated as density) or potential function of the object, not the electron density of the object itself (see the paper by Unwin and Henderson JMB 1975 or the textbook by Glaeser, et al.).

Thanks for noticing this. We changed “electron density” to “density”.

11. Line 171: it states that “The chemical nature of these residues is not essential for trimer formation”. I think it is more appropriate to describe it as a trimeric interface rather than trimer formation, which leaves an implication that trimer formation is the initial/critical step of the assembly. Is there a previous study suggesting whether the inter-dimer within a hexamer or the inter-trimeric among three hexamers is the driving force to initiate the entire virion assembly? It would be interesting to include some discussion on the assembly mechanism of retrovirus which might already be discussed in the literature?

We thank the reviewer for this insight. We have modified the language to clarify that the mutated residues are not essential to the formation of a trimer interface instead of a trimer formation.

There are previous studies on the assembly mechanism of mature HIV-1 capsid, where three assembly models have been proposed, namely a de novo assembly model, a displacement model and a hybrid model (see references below). The immature capsid assembly is more complex, involving two other domains: the MA lattice and the NC domain binding to genomic RNA. The assembly process of immature Gag is still largely unknown.

- Keller PW, Huang RK, England MR, Waki K, Cheng N, Heymann JB, Craven RC, Freed EO, Steven AC. *A two-pronged structural analysis of retroviral maturation indicates that core formation proceeds by a disassembly-reassembly pathway rather than a displacive transition. J Virol. 2013 Dec;87(24):13655-64. doi: 10.1128/JVI.01408-13*

- Frank, G., Narayan, K., Bess, J. et al. *Maturation of the HIV-1 core by a non-diffusional phase transition. Nat Commun 6, 5854 (2015). <https://doi.org/10.1038/ncomms6854>*

- Ning, J., Erdemci-Tandogan, G., Yufenyuy, E. et al. *In vitro* protease cleavage and computer simulations reveal the HIV-1 capsid maturation pathway. *Nat Commun* 7, 13689 (2016).
<https://doi.org/10.1038/ncomms13689>

12. Line 160: The simulations are targeted to characterize the monomer interface either within a hexamer or between hexamers. The authors should clarify in their description which interface is referred to as the intra or inter-protein interface, particularly for the dimer interface.

We thank the reviewer for this suggestion, we changed the text accordingly to specify inter-hexamer interface for the trimer and dimer interfaces and intra-hexamer for the 6HB interactions.

13. Line 163-171: Q118 and R100 in the trimer interface are found to be flexible as measured by RMSF in their simulation and non-essential in the particle assembly in the mutant experiments. This is a well-designed study. Are the densities of these residues less resolved in the map compared to those of other amino acids that are found to be more essential and less flexible?

Indeed, we observe some density for the sidechain of R100 and none for Q118, yet these are not as well defined as the densities we observe for less flexible residues, for instance, Y66 and R97 are very well defined in the density map, we do observe some density for E96, D90 and K85, but it is not as well defined. We have included a visualization of the density for these flexible and less flexible residues in the new Supplementary Fig. 7.

14. Line 192: How was it confirmed that a sodium ion is involved in the interaction between the E132 of neighboring hexamers? Can you detect Na⁺ in the map?

We observed some non-protein densities close to the E132 sidechains between two hexamers. Guided by the MD results, we assume these most likely correspond to Na⁺ ion locations. We have added a visualization of these densities in the new Supplementary Fig. 8.

15. Could you explain in the main text how the plots shown in Suppl. Fig. 6 were generated and how the distances vs flexibility is interpreted ?

We appreciate the reviewer's comments and have added a paragraph in the main text explaining the procedure to calculate residue distances and their relation to flexibility:

"Heavy atom root mean square fluctuations (RMSF) were derived from the simulations to determine the flexibility of amino acids located at multiple CA interfaces. Interfacial residue interactions were also identified by measuring residue to residue distances from the simulation trajectory. Specifically, for salt bridging interactions were characterized by the distance between interacting atoms while for hydrophobic interactions the center-of-mass to center-of-mass distance of the sidechain heavy atoms was measured. Flexibility is a good indicator of the importance and specificity of contacts as more rigid contacts tend to be more essential to stability of an interface, at the same time, more flexible residues tend to have more spread interaction distance distributions indicating fluctuating interactions and lower occupancies."

16. Line 209: h12 is not annotated in Fig. 4a or the figure caption.

H12 is now annotated in Fig. 2c and in the caption of Fig. 4a and Fig. 5a.

17. Line 217: Is there another way to describe this packing as domain-swapping (i.e., NTD-1 interacts with CTD-2 from proteins 1 and 2)? It may be easier to see this in Fig. 4 if there was a panel where each protein was annotated in a different color.

This is an interesting point. Domain-swapping means the interaction (or interface) between domain 1 and domain 2 in molecule 1 becomes the same interaction (or interface) between domain 1 from molecule 1 and domain 2 from molecule 2, and vice versa. This is actually not the case with HIV-1 or HERV-K CA, where NTD-1 does not interact with CTD-1, or at least not in the same way as NTD-1 interacts with CTD-2.

REVIEWERS' COMMENTS

Reviewer #2 (Remarks to the Author):

I have reviewed the revised manuscript and find that the authors were very responsive to my suggestions - most of these focuses on clarifying the evolutionary relationship between HERV-Kcon and other retroviruses, which was originally presented as stages in a lineage rather than as a comparison of sister taxa. To their credit the authors understood and have avoided this all too common mistake. I had no major technical concerns other than better citation of one of the methods, which they have done. Overall this is a very nice study and in combination with work done on capsids of other, exogenous retroviruses, will contribute to a better overall understanding of retroviral particle structure generally and betaretroviruses specifically.

Point-to-point responses to the reviewers' comments

Reviewer #2 (Remarks to the Author):

I have reviewed the revised manuscript and find that the authors were very responsive to my suggestions - most of these focuses on clarifying the evolutionary relationship between HERV-Kcon and other retroviruses, which was originally presented as stages in a lineage rather than as a comparison of sister taxa. To their credit the authors understood and have avoided this all too common mistake. I had no major technical concerns other than better citation of one of the methods, which they have done. Overall this is a very nice study and in combination with work done on capsids of other, exogenous retroviruses, will contribute to a better overall understanding of retroviral particle structure generally and betaretroviruses specifically.

We appreciate the reviewer's positive comments.